# Dietary Wheat Gluten Alters the Gut Microbiome and Plasma Taurine Levels in European Sea Bass (*Dicentrarchus labrax*)

Mary E. M. Larkin  and Allen R. Place *

Institute of Marine and Environmental Technology, University of Maryland Center for Environmental Science, 701 East Pratt St., Baltimore, MD 21202, USA; memlarkin@gmail.com
* Correspondence: place@umces.edu; Tel.: +1-410-234-8831

**Abstract:** Incorporating environmentally sustainable plant-based protein sources into aquafeeds is a priority for the aquaculture industry. Wheat gluten as a plant protein source has been successfully integrated into feeds for several fish species. However, unique nutritional requirements and the potential for intolerances necessitate the evaluation of plant-based diets by species. Previous work by our laboratory indicated that wheat gluten at less than 4% inclusion in a compound feed had a negative impact on growth and survival rate in juvenile cobia (*Rachycentron canadum*). In the current study, we evaluated the effects of wheat gluten inclusion in a plant-based diet in juvenile European sea bass (*Dicentrarchus labrax*). No major differences were observed in terms of growth rate, plasma biochemical parameters, detectable induction of plasma IgM, IgT, or factors capable of binding gliadin in response to the inclusion of 4% dietary wheat gluten. However, plasma levels of taurine doubled in the fish fed wheat gluten, and there were considerable changes to the intestinal microbiome. Despite these measurable changes, the data suggest that dietary inclusion of 4% wheat gluten is well tolerated by juvenile European sea bass in a plant-based feed formulation.

**Keywords:** aquaculture; aquaculture feed; European sea bass; microbiome; taurine; wheat gluten

## 1. Introduction

In the past decade, there has been an abundance of research into alternatives to fishmeal as a primary protein source in aquafeeds. This has stemmed from changing priorities within the aquaculture industry to improve the environmental impact, cost savings, and secure supply of ingredients [1]. Plant proteins derived from soy and wheat have been successfully incorporated into diets for multiple fish species [2,3]. However, previous work by our laboratory indicated that wheat gluten at less than 4% inclusion in a compound feed had a negative impact on growth and survival rate in juvenile cobia (*Rachycentron canadum*) (pending publication).

Wheat gluten, isolated from wheat flour, is used as a protein source and binding agent in feed formulations. In the processing of wheat flour to produce wheat gluten, soluble fibers and starches are removed, and the resulting product contains soluble gliadin and insoluble glutenin proteins [4,5]. In a person with celiac disease, the gliadin proteins of wheat gluten trigger an immune response that causes abdominal stress and potential deficiencies of several amino acids [6–8].

European sea bass are a globally popular aquaculture fish found on restaurant menus as branzino or branzini [9]. A prior study suggested that they performed well on a diet containing wheat gluten (~25%) when fishmeal was also included in the formulation [10]. Another study included 20% gluten in an entirely plant-based formulation and noted differences in growth rates between the fish consuming that diet as compared to a fishmeal-based diet [11]. Both of these studies were performed with older juvenile fish starting at ~190 g. European sea bass, like other teleost fish, have innate and adaptive immunity factors, including IgM and teleost immunoglobulin, IgT [12–14].

Plant-based diets lack taurine, a nutritional component critical for growth and survival that is present in fishmeal. While some fish species have sufficient endogenous synthesis, others require dietary intake [15]. For cobia, another marine carnivore, taurine supplementation to 1.5% was found to be ideal for feed palatability and growth performance [16]. Being uncertain as to the taurine requirements of European sea bass, we supplemented the diets in the study with 1.5% taurine to ensure that taurine deficiency would not be a confounding factor in evaluating the effects of wheat gluten.

In the current study, we sought to determine if the inclusion of 4% wheat gluten into the diet of European sea bass in the early juvenile stage (~25 g) impacted overall growth, health and immune status, and the composition of the intestinal microbiome. We chose a relatively low proportion of wheat gluten since this was the first study to probe its effects in young juveniles and in a fishmeal-free formulation. We anticipated that 4% inclusion could reveal any significant differences between groups while avoiding substantial mortalities. Characterizing outcomes entailed tracking growth rates, assaying plasma parameters and tissue weights, detecting adaptive immune factors IgM and IgT, and evaluating differences in distinct sections of the intestinal microbiome.

## 2. Materials and Methods

### 2.1. Fish System Maintenance and Care

The study was carried out in accordance with the guidelines of the Institutional Animal Care and Use Committee of the University of Maryland Medical School (IACUC protocol #0616014). European sea bass were obtained from the laboratory of Yonathan Zohar, PhD, and maintained in the Aquaculture Research Center (ARC) at the Institute of Marine and Environmental Technology (IMET). Starting at ~25 g of body weight, fish were fed diets containing either 0 or 4% wheat gluten (Zeigler Bros., Gardners, PA, USA), starting with 75 fish per diet. Fish were maintained on these diets for 6 months, at which time the average weight was 250 g. Tank weights were recorded weekly and feeding rates adjusted so that the fish were fed 3.5% of their body weight per day over 3–4 feedings by hand.

Fish were divided by diet and housed in one of 2 eight-foot diameter, four cubic meter, recirculating systems sharing mechanical and biological filtration as well as life support systems. Supplemental aeration was also provided. There was a total of one experimental tank per diet. The recirculating system had a filtration system, which included protein skimming, ozonation, mechanical filtration in the form of bubble-bead filters, and biological filtration. The photoperiod was maintained at 14 h light and 10 h dark for the duration of the experiment.

### 2.2. Diet Preparation

The formulations for the 0 and 4% wheat gluten diets containing 1.5% taurine are shown in Table 1.

**Table 1.** Feed formulations.

| Ingredient (g kg$^{-1}$) | 0% Wheat Gluten | 4% Wheat Gluten |
|---|---|---|
| Profine VF | 28.75 | 26.75 |
| Soybean meal, 47.5% | 23.33 | 23.33 |
| Wheat flour, bagged | 15.04 | 15.04 |
| Corn gluten, 60% | 15.34 | 13.34 |
| Menhaden gold oil, top-dressed | 5.96 | 5.96 |
| Monocalcium phosphate FG | 3.95 | 3.95 |
| Lecithin FG | 3 | 3 |

**Table 1.** *Cont.*

| Ingredient (g kg$^{-1}$) | 0% Wheat Gluten | 4% Wheat Gluten |
|---|---|---|
| L-Lysine, 98.5% | 0.75 | 0.75 |
| Choline chloride, 70% | 0.6 | 0.6 |
| Potassium chloride (DYNA K) FG | 0.56 | 0.56 |
| DL-Methionine, 99% | 0.45 | 0.45 |
| Sodium chloride | 0.28 | 0.28 |
| Tiger C-35 | 0.2 | 0.2 |
| Premix AquaVit | 0.12 | 0.12 |
| Premix Aquamin Fish | 0.12 | 0.12 |
| Magnesium oxide FG | 0.05 | 0.05 |
| Taurine FG, 98.5% | 1.5 | 1.5 |
| Wheat gluten | 0 | 4 |

FG = Food grade.

The proximate composition of the feeds is shown in Table 2. The analysis was performed by New Jersey Feed Laboratory, Inc. (Ewing Township, NJ, USA).

**Table 2.** Proximate composition and measured taurine values.

| Proximate Composition [1] | 0% Wheat Gluten | 4% Wheat Gluten |
|---|---|---|
| Protein (% DM) | 45.02 | 44.94 |
| Lipid (% DM) | 6.19 | 6.02 |
| Fiber (% DM) | 1.95 | 2.2 |
| Moisture (%) | 6.97 | 9.4 |
| Ash (% DM) | 7.75 | 7.58 |
| Taurine (%) | 1.48 | 1.54 |

DM = Dry matter; [1] New Jersey Feed Labs analysis.

### 2.3. Blood and Tissue Sampling and Analysis

At the conclusion of the 6-month trial, food was withheld for 24 h, and 10–12 fish from each diet were anesthetized with 25 mg/L MS-222 (Syndel, Ferndale, WA, USA) buffered with 50 mg/L sodium bicarbonate (Sigma-Aldrich, St. Louis, MO, USA) and exsanguinated via the caudal vein to collect blood for plasma analysis. Following blood collection, the spinal cord was severed, and tissues were harvested for analysis.

Approximately 1 mL blood from the caudal vein was put into a tube containing 20 μL of 1000 units/mL heparin and gently inverted to mix. Samples were centrifuged at 2000× *g* for 15 min at 4 °C, and the plasma fraction retained. This processing was sufficient for plasma chemistry and immunoblotting. In preparation for taurine analysis by high-performance liquid chromatography (HPLC), 10 μL plasma was mixed with 90 μL (1:10) 70% ethanol containing 0.154 mM D-norleucine. The solution was vortexed and centrifuged at 2000× *g* for 5 min. In total, 50 μL supernatant was retained and dried at 70 °C overnight.

### 2.4. Plasma Analysis

Plasma analysis was performed by Jill Arnold of ZooQuatic Laboratory in Baltimore, MD. Samples were processed using standard procedures for biochemistry analytes (Table S1: Standard plasma analytes measured) using the ChemWell-T analyzer (CataChem, Oxford, CT, USA). Calibration and quality control materials were used per manufacturer's instructions (Catacal and Catatrol control level 1 and 2, CataChem, Oxford, CT, USA). Osmolality was measured using the Wescor Osmometer (Wescor, Inc., Logan, UT, USA)

after calibration with two levels of standards (290 and 1000 mmol/kg, OPTIMOLE, ELITech-Group Biomedical Systems, Logan, UT, USA).

### 2.5. Plasma Taurine Analysis by HPLC

Dry samples were resuspended in 300 μL 0.1 N HCl and filtered through 0.45-micron filters (EMD Millipore, Billerica, MA, USA). In total, 5 μL of the filtered extracts were derivatized according to the AccQTag Ultra Derivitization Kit protocol (Waters Corporation, Milford, MA, USA). Amino acids were analyzed using an Agilent 1260 Infinity High-Performance Liquid Chromatography System equipped with ChemStation (Agilent Technologies, Santa Clara, CA, USA) by injecting 5 μL of the derivatization mix onto an AccQTag Amino Acid Analysis C18 (Waters, Milford, MA, USA) 4.0 μm, 3.9 × 150 mm column heated to 37 °C. Amino acids were eluted at 1.0 mL min$^{-1}$ flow with a mix of 10-fold diluted AccQTag Ultra Eluent (C) (Waters Corporation, Milford, MA, USA), ultra-pure water (A) and acetonitrile (B) according to the following gradient: Initial, 98.0% C/2.0% B; 2.0 min, 97.5% C/2.5% B; 25.0 min, 95.0% C/5.0% B; 30.5 min, 94.9% C/5.1% B; 33.0 min, 91.0% C/9.0% B; 38 min, 40.0% A/60.0% B; 43 min, 98.0% C/2.0% B. Derivatized amino acids were detected at 260 nm using a photo diode array detector. Signals were referenced to AABA (alpha-Aminobutyric acid), D-norleucine, and standard hydrolysate amino acids.

### 2.6. Gliadin SDS-PAGE Electrophoresis and Immunoblotting

A solution of gliadin (Sigma, St. Louis, MO, USA) was made to an original concentration of 2 mg/mL and subsequently diluted 2-fold to 1 mg/mL, 0.5 mg/mL, 0.25 mg/mL, and 0.125 mg/mL, all in Laemmli sample buffer. All samples were heated to 95 °C for 3 min and centrifuged at 10,000× *g* for 1 min prior to electrophoresis. Recombinant *Amphidinium carterae* eIF4A-1A, a translation protein, was used as a control for loading, transfer efficiency, and specificity of binding. eIF4E-1A was diluted into Laemmli sample buffer to a concentration of 57.5 ng/μL. In total, 15 μL of a 1:1 volume ratio of gliadin and eIF4A was loaded into each lane of a Novex NuPAGE 4–12% Bis-Tris gel and electrophoresed in a Bolt Mini Gel Tank at 165 V for 1 h with MOPS SDS running buffer (Life Technologies, Frederick, MD, USA).

Poly (vinylidene fluoride) (PVDF) membrane was activated by a brief dip in 100% methanol and equilibrated for 5 min in Novex NuPAGE transfer buffer. The gliadin gel was electroblotted onto prepared PVDF in a Bolt Mini Blot Module at 30 V for 1 h in Novex NuPAGE transfer buffer (Life Technologies, Frederick, MD, USA). Following transfer, the membrane was washed in ddH$_2$0 for 5 min. Duplicate transferred lanes were divided and designated as "0 Wheat Gluten Plasma Block" or "4% Wheat Gluten Plasma Block" for subsequent incubation procedures.

The 0 Wheat Gluten Plasma Block and 4% Wheat Gluten Plasma Block membranes were incubated at room T for 1 h followed by overnight at 4 °C with mixed plasma from 3 fish fed diets containing either 0 wheat gluten or 4% wheat gluten, respectively, diluted 1:12 into TBS-T containing 5% nonfat milk. The following day, blots were washed 4 times for 10 min each time with TBS-T. Both blots were incubated with polyclonal anti-gliadin antibody (Biorbyt, Cambridge, UK) and rabbit anti-*A. carterae* eIF4E-1A (GenScript, Piscataway, NJ, USA) diluted 1:500 and 1:2000, respectively, into TBS-T containing 5% nonfat milk at room temperature for 1 h. The blots were again washed 4 times for 10 min each time with TBS-T. Both blots were incubated with goat anti-rabbit IgG conjugated to horseradish peroxidase (HRP) (Bio-Rad, Hercules, CA, USA) diluted 1:2500 into TBS-T containing 5% nonfat milk at room temperature for 1 h followed by four 10-min washes with TBS-T. The HRP signal from bound antibody was visualized using Clarity Western ECL Substrate (Bio-Rad, Hercules, CA, USA). Imaging was performed in a Flourchem®, and the AlphaView program was used to analyze densitometry (ProteinSimple, San Jose, CA, USA).

### 2.7. IgM and IgT Immunoblotting

Plasma was diluted 1:40 in 1× SDS-PAGE sample buffer. Samples were heated for 3 min at 95 °C and centrifuged for one minute at 10,000× *g*. In total, 13 μL of each sample

was electrophoresed on a 4–12% Bis-Tris protein gel (NuPAGE Novex, ThermoFisher Scientific, Waltham, MA, USA) for 35 min at 200 V using MOPS buffer in a PowerPac HC Power Supply (Bio-Rad, Hercules, CA, USA). Proteins were transferred to a PVDF membrane for 14 min on the high molecular weight setting (25 V) in the Trans-Blot Turbo Transfer System (Bio-Rad, Hercules, CA, USA). Immunoblotting was performed in the iBind Western System (ThermoFisher Scientific, Waltham, MA, USA).

For IgT detection, an anti-European sea bass IgT polyclonal antibody (rabbit IgG "RAIgT1", kindly provided by Giuseppe Scapigliati, Tuscia University, Italy), was used at a dilution of 1:1000 as the primary antibody, and goat anti-rabbit IgG H&L HRP conjugate at a dilution of 1:2000 (Bio-Rad, Hercules, CA, USA) was used as the secondary antibody. For IgM detection, the Magic anti-European sea bass IgM monoclonal antibody (mouse IgG) (Creative Diagnostics, Shirley, NY, USA) was used at a dilution of 1:1000 as the primary antibody, and goat anti-mouse IgG H&L HRP conjugate (Bio-Rad, Hercules, CA, USA) was used as the secondary antibody at a dilution of 1:2000. A chemiluminescent signal was generated with addition of Clarity Western ECL substrate and imaged in a ChemiDoc Touch Imaging System (Bio-Rad, Hercules, CA, USA). Image Lab software (Version 5.2.1, Bio-Rad, Hercules, CA, USA) was used to visualize immunoblots and analyze protein molecular weights.

### 2.8. Microbiome Analysis

#### 2.8.1. DNA Extraction

For sampling of tank water, 1 L of water was filtered through a pore size of 0.2 microns. For the feed, 0.127 g (3 pellets) of each diet was used. For intestinal samples (pyloric caeca, anterior intestine, mid-intestine, and posterior intestine), ~0.25 g tissue was collected. Deoxyribonucleic acid (DNA) extraction was performed using the Qiagen DNeasy Powerlyzer Powersoil kit (Qiagen, Germantown, MD, USA). Samples were manually processed through bead beating twice at 6 m/s for 30 s (FastPrep-24, MP Biomedicals, Santa Ana, CA, USA) and centrifugation for two minutes at $10,000 \times g$ (Eppendorf 5415 D, Sigma-Aldrich, St. Louis, MO, USA). In the case of the water analysis, the filter was treated as the sample for processing. Following Step 5 in the manufacturer's protocol, DNA extraction was completed in a Qiacube (Qiagen, Germantown, MD, USA) according to manufacturer's instructions for DNA extraction, including the optional PCR inhibitor removal. DNA was stored at $-20\,^\circ$C.

#### 2.8.2. PCR and Agarose Gel Electrophoresis

Polymerase chain reaction (PCR) was performed to confirm the presence of amplifiable DNA for 16s rRNA sequencing. One µL of extracted DNA was used in a 25 µL reaction volume with Promega PCR Master Mix (Thermo Fisher Scientific, Waltham, MA, USA) and amplified in a DNA Engine Dyad (MJ Research, Quebec, QC, Canada) using primers 16S_27F 5′ AGAGTTTGATCMTGGCTCAG 3′ and 16S_1492R 5′ TACGGYTACCTTGTTAC-GACTT 3′ with the following reaction conditions:

95 $^\circ$C for 5 min, 92 $^\circ$C for 30 s, 50 $^\circ$C for 2 min, 72 $^\circ$C for 1 min, 30 s, cycle to step 2 for 39 more times, incubate at 72 $^\circ$C for 5 min, hold at 4 $^\circ$C. Agarose gel electrophoresis was performed on each PCR product to detect the presence of a 1465 bp DNA band spanning bacterial rRNA variable regions 1–9. Samples were electrophoresed in a 1% agarose gel containing 0.5 µg/mL ethidium bromide at 150 V for 40 min and imaged in a ChemiDoc Touch Imaging System (Bio-Rad, Hercules, CA, USA).

#### 2.8.3. Sequencing

Sequencing was performed at the BioAnalytical Services Laboratory (BAS Lab) at IMET on an Illumina MiSeq (San Diego, CA, USA) using 5 ng DNA from each sample. Primers complementary to the V3–V4 hypervariable region of the bacterial 16s rRNA gene were designed based on those characterized by Klindworth et al.: S-D-Bact-0341-b-S-17 5′-CCTACGGGNGGCWGCAG-3′ and S-D-Bact-0785-a-A-21 5′-GACTACHVGGGTATCTA ATCC-3′ [17].

Including the Illumina adaptor sequences, the full-length primers were F 5′ TCGTCG-GCAGCGTCAGATGTGTATAAGAGACAGCCTACGGGNGGCWGCAG and R 5′ GTC TCGTGGGCTCGGAGATGTGTATAAGAGACAGGACTACHVGGGTATCTAATCC. CLC Genomics Workbench 8 (Version 8.5.1, Qiagen, Germantown, MD, USA) was used to trim, pair, and merge sequences (default parameters used for all functions). They were exported as a merged FASTA file and imported into the Quantitative Insights into Microbial Ecology (QIIME) program (Version 1.9.1) [18] for open reference operational taxonomic unit (OTU) and taxonomic classification using the Silva 128 reference database [19]. Rarefaction curves were generated based on observed OTUs. Identify threshold was set to 97%. Representative sequence alignments for each OTU were generated using Python Nearest Alignment Space Termination (PyNAST) [20], and R (Version 3.4.3) [21] was used to generate a bar graph of the bacterial orders, as well as a principal component analysis (PCA) plot. QIIME was used to generate the rarefaction plot.

### 2.9. Statistics

Statistical significance was evaluated using analysis of variance (ANOVA) and Student's *t*-test (2-tailed) with a 95% confidence interval (Microsoft Excel, version 16.45).

## 3. Results

### 3.1. Growth Data

Growth rates were equivalent ($p > 0.05$) for the fish fed plant-based diets with or without added wheat gluten (4%). The study commenced with fish of an average weight of ~25 g and continued for 6 months. The growth curve is shown in Figure 1.

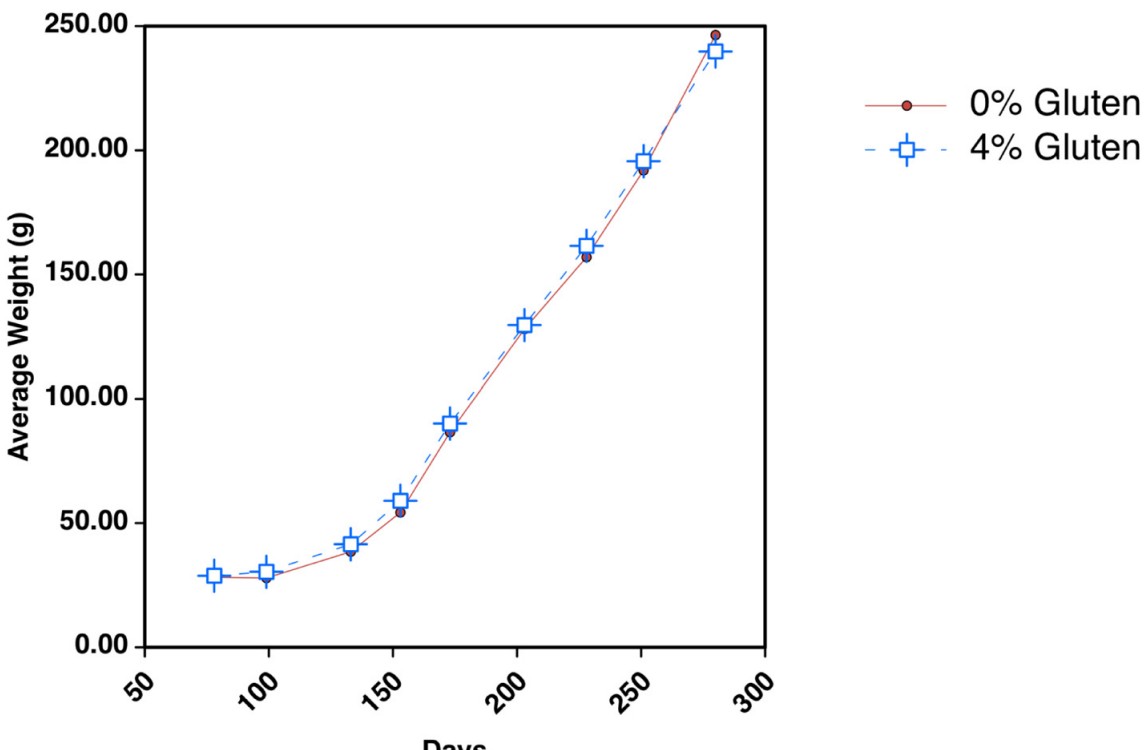

**Figure 1.** Dietary inclusion of 4% wheat gluten does not affect growth. Fish were fed plant-based diets containing 0 or 4% wheat gluten, starting at an average weight of ~25 g. The study continued for 6 months, and growth rates were similar throughout the study ($p > 0.05$).

### 3.2. Plasma Analysis

As shown in Table 3, analysis of common plasma analytes revealed no differences in levels between the 0 or 4% wheat gluten dietary groups with the exception of calcium and

aspartate aminotransferase (AST) ($p < 0.05$). Plasma calcium is higher in the group fed 4% wheat gluten, whereas plasma AST levels are lower.

**Table 3.** Levels of common plasma parameters for fish fed diets with 0 or 4% wheat gluten are similar in value except for levels of calcium and AST.

| Plasma Component | 0% Wheat Gluten | 4% Wheat Gluten |
|---|---|---|
| Glucose (mg/dL) | 120.727 ± 27.335 | 113.874 ± 32.011 |
| Calcium (mg/dL) | * 11.045 ± 0.465 | * 11.75 ± 0.567 |
| Magnesium (mg/dL) | 3.545 ± 0.398 | 3.7 ± 0.25 |
| Phosphorus (mg/dL) | 7.909 ± 0.76 | 8.125 ± 0.784 |
| Triglycerides (mg/dL) | 410.364 ± 77.747 | 388 ± 129.836 |
| Cholesterol (mg/dL) | 199.545 ± 33.074 | 206.75 ± 43.702 |
| ALP (U/L) | 24.091 ± 1.486 | 25.125 ± 1.511 |
| AST (U/L) | * 47.8 ± 28.558 | * 35.25 ± 17.484 |
| Osmolality (mOsmol/kg) | 363 ± 5.773 | 368.917 ± 6.851 |
| Total Protein (g/dL) | 4.291 ± 0.241 | 4 ± 0.525 |

* Statistically different between diets; ALP = alkaline phosphatase; AST = aspartate aminotransferase.

### 3.3. Body and Tissue Weights

Whole body and tissue weights are similar between fish fed either 0 or 4% wheat gluten with the exception of the mid-intestine ($p < 0.05$). These data are summarized in Table 4. Hepatosomatic indices were an average of 0.01 for both dietary groups.

**Table 4.** Body and tissue weights for fish fed 0 or 4% wheat gluten are similar with differences manifesting in the mid-intestine.

| Weight (g) | 0% Wheat Gluten | 4% Wheat Gluten |
|---|---|---|
| Whole body | 312.142 ± 68.727 | 311.667 ± 71.247 |
| Pyloric caeca | 0.818 ± 0.402 | 0.805 ± 0.266 |
| Anterior intestine | 0.875 ± 0.194 | 1.03 ± 0.235 |
| Middle intestine | * 0.576 ± 0.13 | * 0.932 ± 0.335 |
| Posterior intestine | 0.621 ± 0.16 | 0.753 ± 0.238 |
| Total intestine | 2.89 ± 0.605 | 3.52 ± 0.918 |
| Liver | 3.625 ± 1.188 | 3.703 ± 1.064 |

DM = Dry matter; * Statistically different between diets.

### 3.4. Plasma Taurine Analysis

Though there were minimal differences between levels of most common plasma analytes, a separate analysis of plasma taurine levels by HPLC showed that levels were approximately twice as high in the fish fed the diet containing wheat gluten (Figure 2).

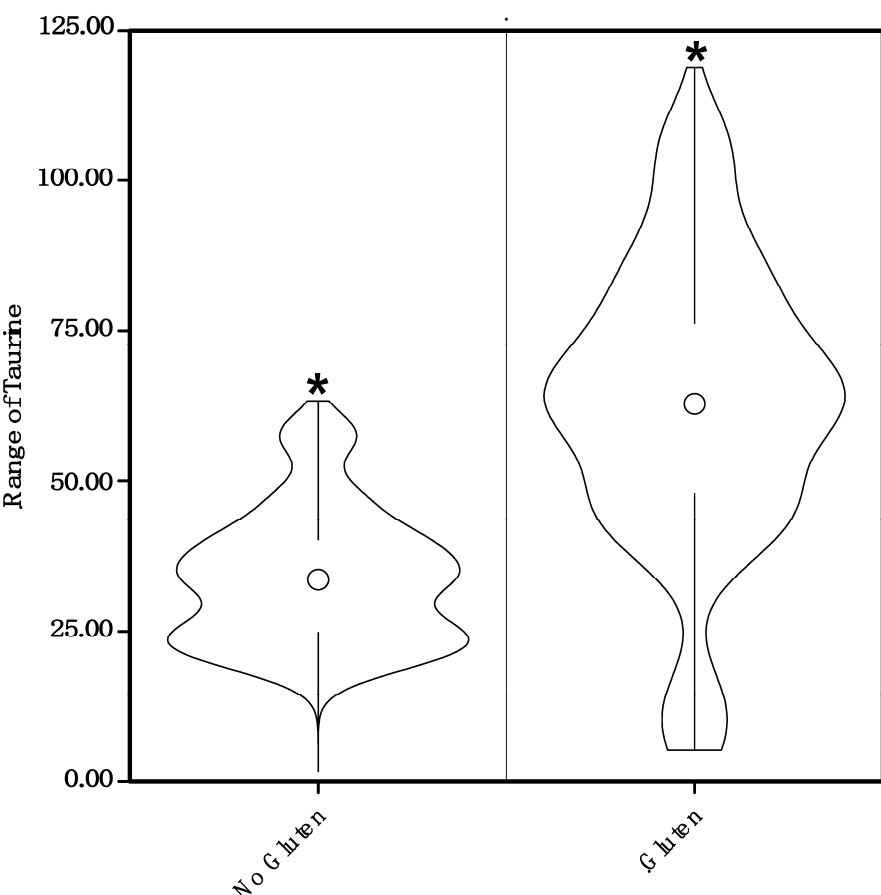

**Figure 2.** The 4% dietary wheat gluten substantially raises plasma taurine levels. Taurine levels in plasma were measured using HPLC. Average concentrations of plasma taurine for fish fed the 0 or 4% wheat gluten diets are 33.82 ± 7.133 or 62.515 ± 15.719 nmol/mL, respectively. * Statistically different between diets.

### 3.5. Gliadin Immunoblotting

Another plant-based aquafeed study performed by our laboratory revealed that juvenile cobia produced a plasma factor capable of binding gliadin when they were fed a diet containing 3.2–3.6% wheat gluten (pending publication). To detect plasma factors capable of binding to gliadin in European sea bass, varying concentrations of the protein as well as a fixed amount of eIF4E-1A from the dinoflagellate *A. carterae,* were subjected to immunoblotting. After gliadin transfer, membranes were pre-incubated in blocking buffer containing mixed plasma from 3 fish fed either the 0 or 4% wheat gluten diet. In the immunoblot, bands are visible in the 30–40 kDa range corresponding to various α/β and γ gliadins (Figure 3) [22]. These gliadin bands are less visible over the course of 2-fold dilutions. There does not appear to be a component in European sea bass plasma produced in response to dietary wheat gluten that is capable of binding gliadin. If binding occurred, there would be a reduced signal resulting from decreased binding of the primary antibody and subsequently diminished binding of the signal-conjugated secondary antibody.

### 3.6. IgT and IgM

We used immunoblotting to assay for IgT and IgM to see if the dietary inclusion of wheat gluten alters their levels in plasma. The results shown in Figure 4 suggest that levels do not change in response to dietary wheat gluten. The size of ~73 kDa for the heavy chains of the antibodies is similar to the size of ~78 kDa detected by Picchietti et al. [23].

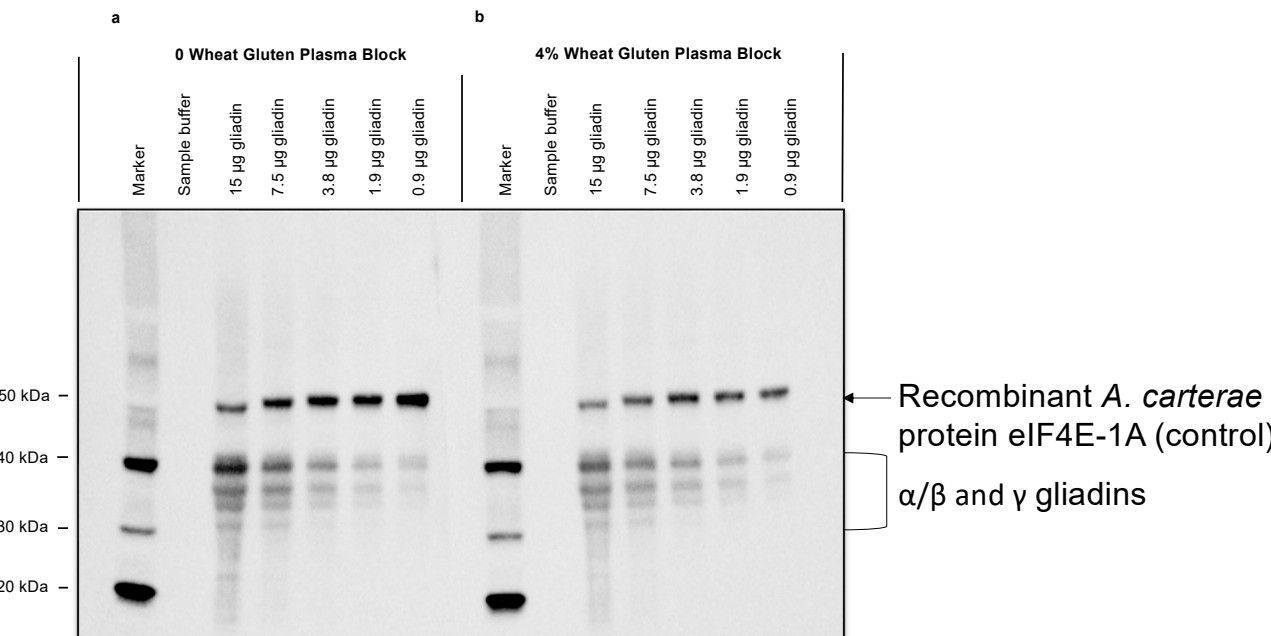

**Figure 3.** The induction of plasma factors capable of binding gliadin is not detectable in fish fed a 4% wheat gluten diet. Immunoblotting was performed with gliadin protein and mixed plasma extracts from fish fed either the 0 wheat gluten diet (**a**) or the 4% wheat gluten diet (**b**). No factors capable of binding to gliadin (30–40 kDa) and inhibiting the binding of anti-gliadin polyclonal antibodies are detectable. Each lane also contains 431 ng recombinant *A. carterae* protein eIF4E-1A (50 kDa) as a control for loading, transfer efficiency, and specificity of binding.

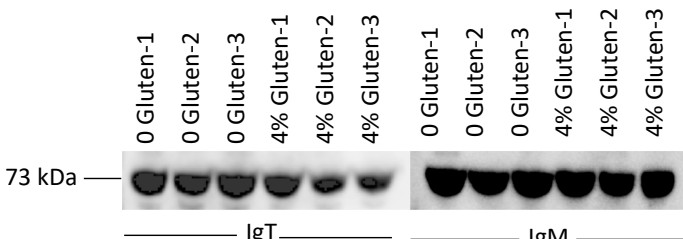

**Figure 4.** Dietary wheat gluten does not induce changes to levels of plasma IgT or IgM. IgT and IgM were measured using immunoblotting, and their levels are not significantly altered in response to 4% dietary wheat gluten.

### 3.7. Intestinal Microbiome Analysis

To characterize the microbiome of the water, feed, and intestinal sections, 16S rRNA gene analysis was performed using the MiSeq platform. Figure 5 shows a bar graph of order-level taxonomic abundance. Each bar represents the analysis of a single fish. Proteobacteria dominate in the tank water. The "cyanobacteria" in the feed are most likely chloroplasts from plant ingredients. This is also true for "cyanobacteria" in the pyloric caeca, which likely corresponds to undigested feed despite the fact that food was withheld for 24 h before tissue sampling. For fish consuming no dietary gluten (Tank 11), the predominant phylum for all intestinal sections analyzed (pyloric caeca (PC), anterior intestine (A), mid-intestine (M), and posterior intestine (P)) is Proteobacteria. The same is true for the intestinal sections of the fish fed 4% gluten (Tank 12), but there is a greater diversity of predominant orders of Proteobacteria and Bacteroidetes present in the mid- and posterior intestines.

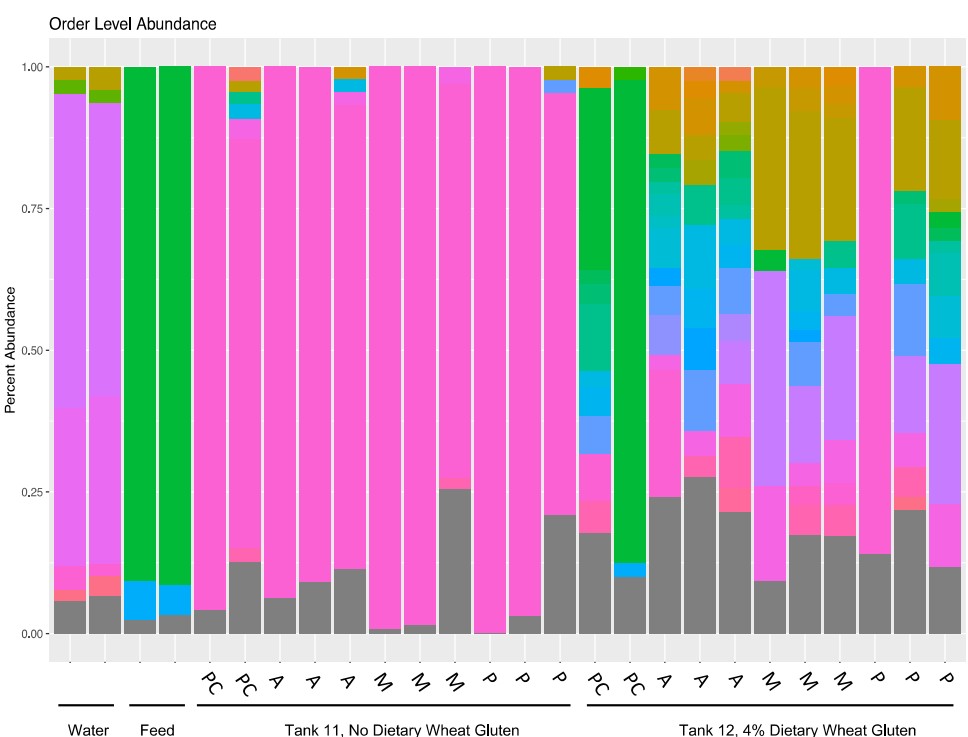

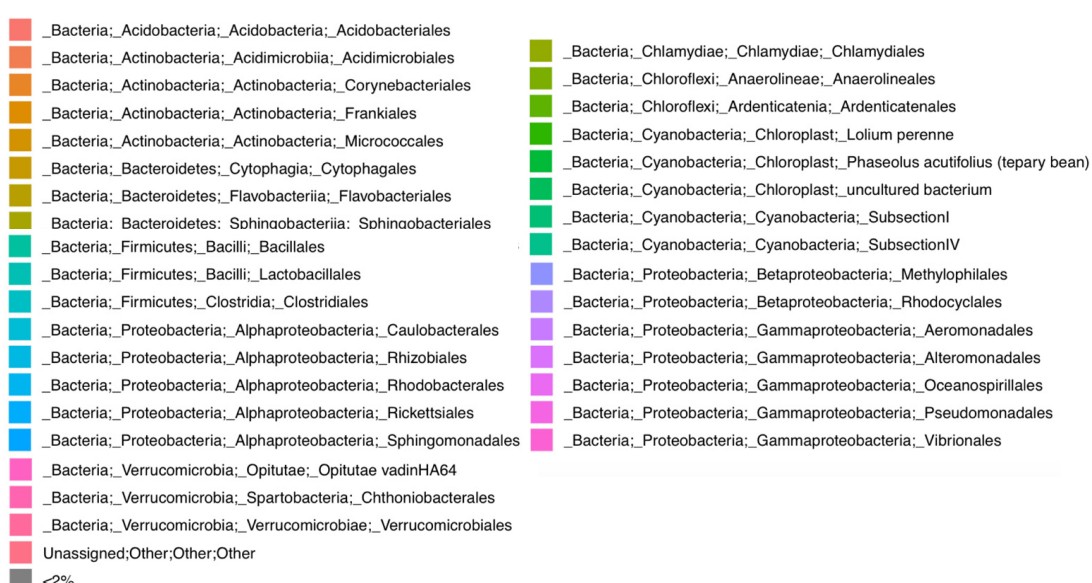

**Figure 5.** Fish fed 4% wheat gluten as part of a plant-based diet exhibit a greater diversity of predominant taxonomic orders across the intestine. DNA extracted from water, feed, and various sections of the intestine from individual fish for the two diet groups underwent 16S rRNA gene analysis using the MiSeq platform to characterize the microbial landscape. The addition of 4% wheat gluten to a plant-based diet dramatically shifted the intestinal microbiome. Intestinal sections analyzed: pyloric caeca (PC), anterior intestine (A), mid-intestine (M), and posterior intestine (P).

Figure 6 shows an alpha-diversity rarefaction curve based on operational taxonomic unit (OTU) data. The principal component analysis (PCA) plot in Figure 7 shows that samples primarily cluster by the absence or presence of dietary wheat gluten. The ellipse shows a 95% confidence interval.

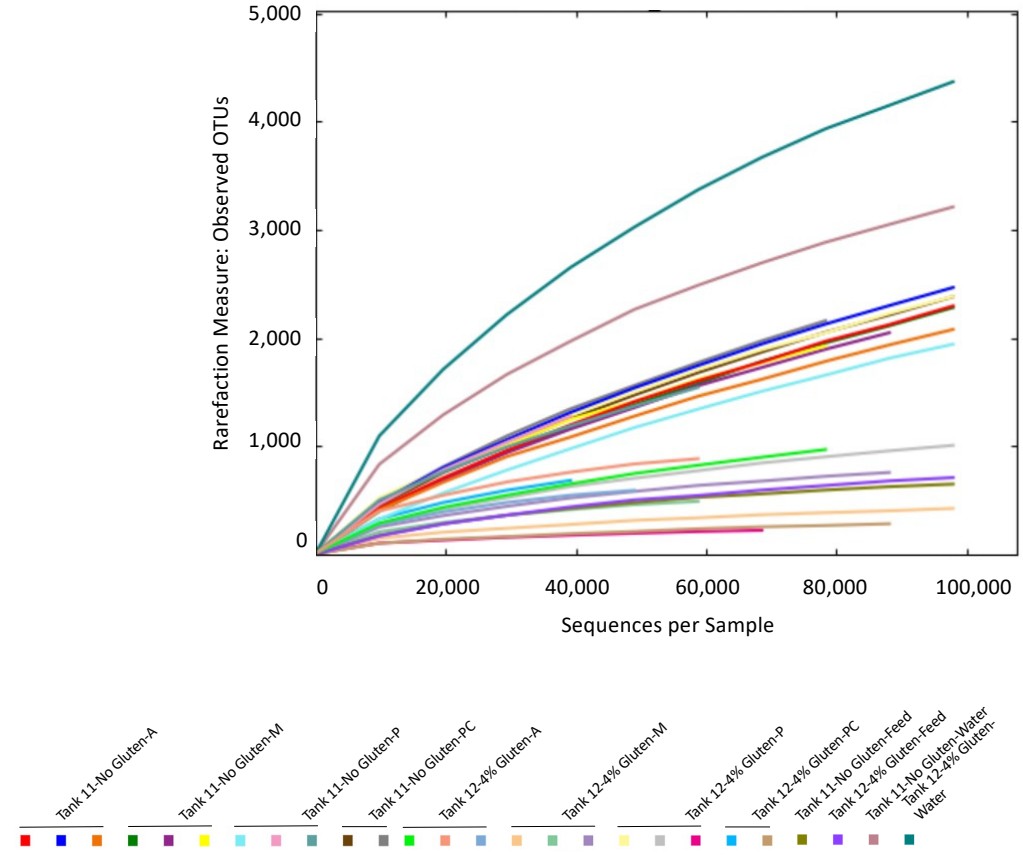

**Figure 6.** Species richness for each microbiome from individual fish, feed, or water samples. The species richness of the microbiome samples is shown in a rarefaction curve of observed OTUs vs. sequences per sample.

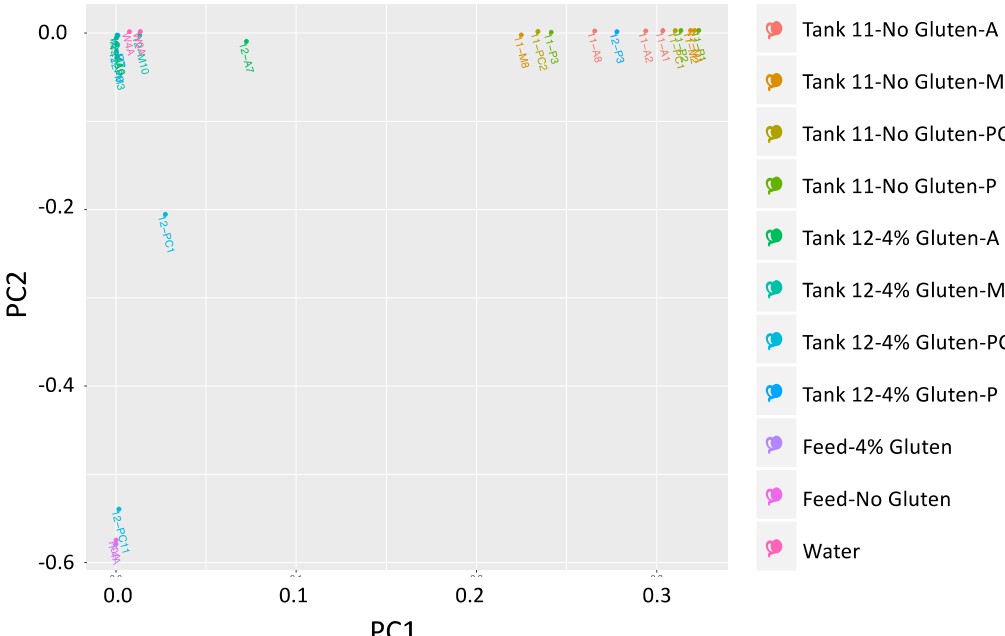

**Figure 7.** Individual fish samples primarily cluster by the absence or presence of dietary wheat gluten in a PCA plot. The ellipse shows a 95% confidence interval. Intestinal sections analyzed: pyloric caeca (PC), anterior intestine (A), mid-intestine (M), and posterior intestine (P).

## 4. Discussion

There were no significant differences in growth between the 0 and 4% wheat gluten cohorts, the primary measurement of a successful diet when raising fish for food. The lack of detectable induction of higher levels of plasma IgM and IgT suggests a lack of induced adaptive immune response to wheat gluten. We also attempted to detect TNF-α in the plasma samples but obtained a high standard error, suggesting that our antibody was not specific. There was no detectable plasma factor capable of binding to gliadin.

The common plasma measures of overall health that differed between the 0 and 4% groups were calcium and AST. Plasma calcium was higher in the group fed 4% wheat gluten, whereas plasma AST levels were lower than they were in the fish fed the diet without wheat gluten. European sea bass stressed by hydrogen peroxide exposure can exhibit higher levels of plasma calcium, so it could be indicative of a health effect [24]. AST is an indicator of liver health and function, along with ALP (alkaline phosphatase) [25]. In a fasting study on European sea bass performed by Peres et al., AST levels increased during the fasting period [26]. Our values for all plasma markers measured are similar to those reported by Peres et al. in European sea bass maintained on a fishmeal diet with the exception of cholesterol, ALP, and AST. For ALP and AST, our values were substantially lower, almost half of the values obtained by their group. This may be a function of a plant-based vs. fishmeal-based diet. Changes to cholesterol were not apparent in our study, but they have been in other studies with wheat gluten as a feed component. Plasma cholesterol decreased in European sea bass fed diets with graded levels of wheat gluten partially replacing up to 70% of the fishmeal as a protein source, but that may be due to the greater proportion of plant protein incorporation rather than an effect unique to wheat gluten [27]. Interestingly, a partial replacement of fishmeal with corn as a protein source and no added wheat gluten was shown to raise plasma cholesterol and phospholipid levels. The authors attributed this effect to the higher carbohydrate content in the diet containing corn [28]. Different plant sources of protein may have a variety of influences on plasma parameters.

The only change in measured tissue weights between diets was for the mid-intestine. There have been no reports of major histological changes to only the mid-intestine prompted by dietary ingredients. However, a gene expression study performed in European sea bass suggested that there is functional specialization across the length of the intestinal tract [29]. Another study found that mid-intestine lactic acid bacteria (order *Lactobacillales*) are highly modulated by diets in a recirculating aquaculture system [30].

There are no studies correlating wheat gluten to higher levels of taurine in plasma. Recently, there have been reports of diets for pets containing legumes marketed as "grain-free," causing cardiomyopathies related to taurine deficiency [31]. It is possible that a feed ingredient such as wheat gluten might be influencing the sequestration of taurine in plasma. Alternatively, endogenous levels of taurine may be increased as a result of dietary wheat gluten, potentially to counter some pro-inflammatory effects of the wheat gluten.

The dietary inclusion of wheat gluten induced marked changes to the intestinal microbiome. In the intestines of fish fed 4% wheat gluten, there were increased predominant diversities of Proteobacteria as compared to the 0 wheat gluten group as well as the presence of Bacteroidetes in the mid- and posterior intestines. Several studies have linked dietary changes to alterations in the intestinal microbiome, and a small number have probed for this effect, specifically with wheat gluten. Human studies in which participants shifted to a gluten-free diet showed variations in the microbiome, though the most significant variation was inter-patient. One of these studies found a significant decrease in the family *Veillonellaceae* of the *class Clostridia* [32,33]. A study in zebrafish demonstrated that fish fed diets containing wheat gluten (~50% of formulation) had heightened abundances of *Legionellales*, *Rhizobiaceae*, and *Rhodobacter* over fishmeal-fed fish [34]. In another study on zebrafish, fish fed wheat gluten had decreased *Bifidobacterium* relative to fish fed brine shrimp [35]. Only an abstract could be located for this study, so the actual percentage of wheat gluten is unknown. In a study of Atlantic salmon, wheat gluten (~14–20% of formulation) mixed with a legume protein (soybean meal or guar meal) increased the

abundance of lactic acid bacteria in the gut as compared to the reference fishmeal diet [36]. Our data do not appear to correlate with these other studies, but many of the differences are likely attributable to the difference between species and the overall constitution of the diet: plant-based, fishmeal-based, or a combination of both. Overall, the study results do not seem to indicate any type of disease state induced by the addition of 4% wheat gluten, as the fish have overall health comparable to that of the fish consuming a diet containing no wheat gluten.

Preliminary data from another study performed by our laboratory, in which European sea bass were fed diets containing 0 or 5% taurine, suggest that European sea bass are adequate endogenous synthesizers of taurine and do not require supplementation. It is possible that the ability to synthesize taurine is related to wheat gluten tolerance. Cobia are inadequate synthesizers of taurine and have a low tolerance for wheat gluten in the juvenile stage of development (pending publication). Future studies of the effects of dietary wheat gluten in other marine carnivorous species with various taurine requirements may clarify this connection.

**Supplementary Materials:** The following supporting information can be downloaded at https://www.mdpi.com/article/10.3390/jmse11051022/s1, Table S1: Standard plasma analyzes measured.

**Author Contributions:** Conceptualization, M.E.M.L. and A.R.P.; methodology, M.E.M.L. and A.R.P.; software, M.E.M.L. and A.R.P.; validation, M.E.M.L.; formal analysis, M.E.M.L.; investigation, M.E.M.L.; resources, M.E.M.L. and A.R.P.; data curation, M.E.M.L.; writing—original draft preparation, M.E.M.L.; writing—review and editing, M.E.M.L. and A.R.P.; visualization, M.E.M.L. and A.R.P.; supervision, A.R.P.; project administration, A.R.P.; funding acquisition, A.R.P. All authors have read and agreed to the published version of the manuscript.

**Funding:** This work is supported in part by the USDA National Institute of Food and Agriculture through the Agriculture and Food Research Initiative SAS program (award #2021-68012-35922) and by NOAA/National Sea Grant (award #NA18OAR4170070).

**Institutional Review Board Statement:** All experiments involving fish were carried out in accordance with the guidelines of the Institutional Animal Care and Use Committee of the University of Maryland Medical School: IACUC protocol #0616014. Fish used for tissue sampling were anesthetized with Tricaine methanosulfonate (MS-222, 70 mg $L^{-1}$) for blood sampling and then euthanized with MS-222 (150 mg $L^{-1}$).

**Informed Consent Statement:** Not applicable.

**Data Availability Statement:** Additional data supporting reported results can be provided by authors upon request.

**Acknowledgments:** The authors would like to thank current and former staff of the Aquaculture Research Center at the Institute of Marine and Environmental Technology: Steve Rodgers and Chris Tollini. The authors would also like to thank John Stubblefield of the laboratory of Yonathan Zohar, for assistance with fish sampling and Ernest Williams, formerly of the laboratory of Allen Place, for assistance with R software analysis. This is contribution #23-102 from IMET.

**Conflicts of Interest:** The authors declare no conflict of interest.

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
