# Peer review of "Dietary Wheat Gluten Alters the Gut Microbiome and Plasma Taurine Levels in European Sea Bass (Dicentrarchus labrax)"

_jmse, doi:10.3390/jmse11051022_

Round 1

Reviewer 1 Report

Comments and Suggestions for Authors

In this study Larkin et al investigates the effects of dietary supplementation of wheat gluten on growth, immune marker induction, microbiome composition and taurine production in Sea bass. While the data looks good, I have some comments to address before making a decision on publication.

1. The rationale for using 4% gluten in this study have to be explained somewhere within the manuscript, especially given that previous studies such as the referenced article used 20% gluten for supplementation. How do you know 4% is sufficient to observe any significant differences or that this amount that is used in the tested parameters is biologically relevant, ie close to the amount fish receives in natural environments?

2. Some of the figures are very pixelated and of low resolution, please use higher quality figures for clarity.

3. It is mentioned in lines 331-332 that the IgT and IgM levels do not change in response to dietary wheat gluten. However IgT levels look lower in 4% gluten supplementation group visually, have you quantified the protein secretion based on the band intensity using software like ImageJ? What is the basis of confirming that there is no change in IgT levels. 

4. The figure 6 has to be revised or explained better, which is the control group and which is the 4% gluten supplemented group is not clear, and what is 6-11, 6-12, why only some fish numbers exist such as fish 1, 2, 3, 7, 8 and 11, are not clear? Please expand or revise the figure to improve clarity.

5. As per Table 1, Profine VF quantity is different between 0% and 4% wheat gluten supplementary diets (28.75 and 26.75), the observed effects could also be a result of the difference in Profine VF between the two diets, and should be discussed.

Reviewer 2 Report

Comments and Suggestions for Authors

The introduction is written to indicate that sea bass have a dietary fishmeal requirement. If sea bass can tolerate up to 25% wheat gluten, why include growth in the study?  The introduction needs to be reorganized to be more cohesive and logical. 

List acronyms when they are first used.

Lines 44-46:  Is the wheat gluten the only difference or is the fishmeal diet and plant ingredient diet completely different?  Did this study look at graded level of wheat gluten or is that not the focus of the study.  This needs to be clarified. 

Lines 46-48:  This should be moved to the end of the introduction.  While there is more background after this sentence.

Line 198:  Were the individual fish sample kept separate or pooled for each tank?

Figure 5 legend needs to be clarified.  Are the numbers individual fish or tanks? 

Figure 7:  THE samples are clustered by taurine, feed and water but the legend says wheat gluten.  Please clarify, is taurine the driving force behind the differences in microbial communities?

Lines 382-383:  Are the plasma calcium levels biologically different or just statistically different? Same for AST.

Line425-426:  Provide the reference for this sentence to clarify what study you are referring to.
